# An Efficient Adversarial Attack for Tree Ensembles

**Chong Zhang**     **Huan Zhang**     **Cho-Jui Hsieh**
Department of Computer Science, UCLA
`chongz@cs.ucla.edu, huan@huan-zhang.com, chohsieh@cs.ucla.edu`

## Abstract

We study the problem of efficient adversarial attacks on tree based ensembles such as gradient boosting decision trees (GBDTs) and random forests (RFs). Since these models are non-continuous step functions and gradient does not exist, most existing efficient adversarial attacks are not applicable. Although decision-based black-box attacks can be applied, they cannot utilize the special structure of trees. In our work, we transform the attack problem into a discrete search problem specially designed for tree ensembles, where the goal is to find a valid "leaf tuple" that leads to mis-classification while having the shortest distance to the original input. With this formulation, we show that a simple yet effective greedy algorithm can be applied to iteratively optimize the adversarial example by moving the leaf tuple to its neighborhood within hamming distance 1. Experimental results on several large GBDT and RF models with up to hundreds of trees demonstrate that our method can be thousands of times faster than the previous mixed-integer linear programming (MILP) based approach, while also providing smaller (better) adversarial examples than decision-based black-box attacks on general $\ell_p$ ($p = 1, 2, \infty$) norm perturbations. Our code is available at `https://github.com/chong-z/tree-ensemble-attack`.

## 1 Introduction

It has been widely studied that machine learning models are vulnerable to adversarial examples (Szegedy et al., 2013; Goodfellow et al., 2015; Athalye et al., 2018), where a small imperceptible perturbation on the input can easily alter the prediction of a model. A series of adversarial attack methods have been proposed on continuous models such as neural networks, which can be generally split into two types. The gradient based methods formulate the attack into an optimization problem on a specially designed loss function for attacks, where the gradient can be acquired through either back-propagation in the white-box setting (Carlini, Wagner, 2017; Madry et al., 2018), or numerical estimation in the soft-label black-box setting (Chen et al., 2017; Tu et al., 2018; Ilyas et al., 2018). The decision based (or hard-label black-box) methods only have access to the output label, which usually starts with an initial adversarial example and minimizes the perturbation along the decision boundary (Brendel et al., 2018; Brunner et al., 2018; Cheng et al., 2019, 2020; Chen et al., 2019c).

In this paper we study the problem of efficient adversarial attack on tree based ensembles such as gradient boosting decision trees (GBDT) and random forests (RFs), which have been widely used in practice (Chen, Guestrin, 2016; Ke et al., 2017; Zhang et al., 2017; Prokhorenkova et al., 2018). We minimize the perturbation to find the *smallest possible* attack, to uncover the true weakness of a model. Different from neural networks, tree based ensembles are non-continuous step functions and existing gradient based methods are not applicable. Decision based methods can be applied but they usually require a large number of queries and may easily fall into local optimum due to rugged decision boundary. In general, finding the exact minimal adversarial perturbation for tree ensembles is NP-complete (Kantchelian et al., 2015), and a feasible approximation solution is necessary to evaluate the robustness of large ensembles.

The major difficulty of attacking tree ensembles is that the prediction remains unchanged within regions on the input space, where the region could be large and makes continuous updates inefficient. To overcome this difficulty, we transform the continuous $\mathbb{R}^d$ input space into a discrete $\{1, 2, \ldots, N\}^K$ "leaf tuple" space, where $N$ is the number of leaves per tree and $K$ is the number of trees. On the leaf tuple space we define the distance between two input examples to be the number of trees that have different prediction leaves (i.e., hamming distance), and define the neighborhood of a tuple to be all valid tuples within a small hamming distance. In practice, we propose the attack that iteratively optimizes the adversarial leaf tuple by moving it to the best adversarial tuple within the neighborhood of distance 1. Intuitively we could reach a far away adversarial tuple through a series of smaller updates, based on the fact that each tree makes prediction independently.

In experiments, we compare $\ell_{1,2,\infty}$ norm perturbation metrics across 10 datasets, and show that our method is thousands of times faster than MILP (Kantchelian et al., 2015) on most of the large ensembles, and 3~72x faster than decision based and empirical attacks on all datasets while achieving a smaller distortion. For instance, with the standard (natural) GBDT on the MNIST dataset with 10 classes and 200 trees per class, our method finds the adversarial example with only 2.07 times larger $\ell_\infty$ perturbation than the optimal solution produced by MILP and only uses 0.237 seconds per test example, whereas MILP requires 375 seconds. As for other approximate attacks, SignOPT (Cheng et al., 2020) finds a 13.93 times larger $\ell_\infty$ perturbation (compared to MILP) using 3.7 seconds, HSJA (Chen et al., 2019c) achieves a 8.36 times larger $\ell_\infty$ perturbation using 1.8 seconds, and Cube (Andriushchenko, Hein, 2019) achieves a 4 times larger $\ell_\infty$ perturbation using 4.42 seconds. Additionally, although $\ell_p$ distance is widely used in previous attacks and a small $\ell_p$ perturbation is usually invisible, our method is general and can also be adapted to other distance metrics.

## 2  Background and Related Work

**Problem Setting**    While the main idea can be applied to multi-class classification models and targeted attacks, for simplicity we consider a binary classification model $f : \mathbb{R}^d \to \{-1, 1\}$ consisting of $K$ decision trees. Each tree $t$ is a weak learner $f_t : \mathbb{R}^d \to \mathbb{R}$ of $N$ leaves, and the ensemble returns the sign $f(x) = \text{sign}(\sum_{t=1}^{K} f_t(x))$. Given a victim input example $x_0$ with $y_0 = f(x_0)$, we want to find the **minimal adversarial perturbation** $r_p^*$, determining the **adversarial robustness** under $\ell_p$ norm:

$$r_p^* = \min_{\delta} \|\delta\|_p \quad \text{s.t.} \quad f(x_0 + \delta) \neq y_0. \tag{1}$$

**Exact Solutions**    In general computing the exact (optimal) solution for Eq. (1) requires exponential time: Kantchelian et al. (2015) showed that the problem is NP-complete for general ensembles and proposed a MILP based method; On the other hand, faster algorithms exist for models of special form: Zhang et al. (2020) restricted both the input and prediction of every tree $t$ to binary values $f_t : \{-1, 1\}^d \to \{-1, 1\}$ and provided an integer linear program (ILP) based formulation about 4 times faster than Kantchelian et al. (2015); Andriushchenko, Hein (2019) showed that the exact robustness of boosted decision stumps (i.e., *depth* $= 1$) can be solved in polynomial time; Chen et al. (2019b) proposed a polynomial time algorithm to solve a single decision tree.

**Approximate Solutions**    To get a feasible solution for general models a series of methods have been proposed to compute the *lower bound* (**robustness verification**) and the *upper bound* (**adversarial attacks**) of $r_p^*$. Chen et al. (2019b) formulated the robustness verification problem into a max-clique problem on a multi-partite graph and produced the *lower bound* on $\ell_\infty$; Wang et al. (2020) extended the verification method and produced a *lower bound* on general $\ell_p$ norms; Lee et al. (2019) verified the $\ell_0$ robustness on the randomly smoothed ensembles which is not directly related to our work. On the other hand, decision based attacks (Brendel et al., 2018; Brunner et al., 2018; Cheng et al., 2019, 2020; Chen et al., 2019c) can be applied here to produce an *upper bound* since they don't have architecture or smoothness assumptions on $f(\cdot)$, however they are usually ineffective due to the discrete nature of tree models; Andriushchenko, Hein (2019) proposed the Cube attack for tree ensembles that does stochastic updates along the $\ell_\infty$ boundary, which typically achieves better results than decision based attacks; Yang et al. (2019) focused on the theoretical analysis of search space decomposition, and proposed RBA-Appr to search over a subset of the $N^K$ convex polyhedrons containing training examples; Zhang et al. (2020) restricted both the input and prediction of all trees to binary values and provided a heuristic attack on $\ell_0$ by assigning empirical weights to each feature, in both the white-box setting and a special "black-box" setting via training substitute models. In contrast,

Table 1: Key differences to prior adversarial attacks that are applicable to general tree ensembles.

| | SignOPT | HSJA | Cube | RBA-Appr | Ours |
|---|---|---|---|---|---|
| Access Level | black-box | | black-box | white-box + data | white-box |
| Search Space | input space | | input space | training data | leaf tuple space |
| Step Size | small steps in continuous space | | $\ell_0$ boundary | N/A | one leaf node |
| Model Queries / iteration | 200 | 100~632 | 100 | N/A | ~1 (see §3.4.1) |

our method works on ensemble of general trees $f_t : \mathbb{R}^d \to \mathbb{R}$, and utilizes the special properties of tree models to produce a tighter *upper bound* on general $\ell_p$ ($p = 1, 2, \infty$) norms efficiently. Table 1 highlights our key differences to prior adversarial attacks that are applicable to general tree ensembles.

**Robust Training**    To overcome the vulnerability, Kantchelian et al. (2015) proposed adversarial boosting by appending adversarial examples to the training dataset; Chen et al. (2019a) optimized the worst case perturbation through a max-min saddle point problem and effectively increased the minimal adversarial perturbation; Andriushchenko, Hein (2019) upper bounded the robust test error by the sum of the max loss of each tree, and proposed a training scheme by minimizing this upper bound; Recently Chen et al. (2019d) approximated the saddle point objective with a greedy heuristic algorithm and further increased the robustness. We consider both the standard (natural) models and the robustly trained models (Chen et al., 2019a) to demonstrate our performance on different settings.

## 3   Proposed Algorithm

We propose an iterative approach where the algorithm starts with an initial adversarial example $x'$ s.t. $f(x') \neq y_0$ and greedily moves closer to $x_0$. At each iteration we choose a new adversarial example $x'_{\text{new}}$ within a small *neighborhood* around $x'$ that has the minimum $\ell_p$ distance to $x_0$. Formally we define the update rule below, and the algorithm stops if $x'_{\text{new}}$ does not give smaller perturbation than $x'$. The key problem is to define the neighborhood so that Eq. (2) can be efficiently solved, and we provide an efficient formulation in following sections. We defer all the proofs to Appendix C.

$$x'_{\text{new}} = \underset{x}{\arg\min} \|x - x_0\|_p \quad \text{s.t.} \quad x \in \text{Neighbor}(x'), \ f(x) \neq y_0. \tag{2}$$

### 3.1   Transform Continuous Input Space into Discrete Leaf Tuples

In most of the existing attacks, Eq. (2) is solved by a continuous optimization algorithm where the $\text{Neighbor}(x')$ (the region where we find an improved solution) is a small $\ell_p$ ball around the current solution $x'$. The major difficulty of attacking tree ensemble is that the model prediction will remain unchanged within a region (which may not be small) containing $x'$, so traditional continuous distance measurements are not suitable here. And if we define the neighborhood as a large $\ell_p$ ball, due to the non-continuity of trees, $f(x)$ becomes intractable to enumerate. To handle these difficulties we introduce the concept of leaf tuple and rewrite Eq. (2) into a discrete form. Given an input example $x = [x_1, \dots, x_d]$ the traverse starts from the root node of each tree. Each internal node of index $i$ has two children and a feature split threshold $(j^i, \eta^i)$, and $x$ will be passed to the left child if $x_{j^i} \leq \eta^i$ and to the right child otherwise. The leaf node has a prediction label $v^i$, and will output $v^i$ when $x$ reaches here. We use $\mathcal{C}(x) = (i^{(1)}(x), \dots, i^{(K)}(x))$ to denote the index tuple of $K$ prediction leaves from input $x$. In general we use the subscript $\cdot_j$ to denote the $j_{\text{th}}$ dimension, and the superscripts $\cdot^i$ or $\cdot^{(t)}$ to denote the $i_{\text{th}}$ node and $t_{th}$ tree respectively.

**Definition 1.** (*Bounding Box*) $B^i = (l_1^i, r_1^i] \times \cdots \times (l_d^i, r_d^i]$ denotes the bounding box of node $i$, which is the region that $x$ will fall into this node following the feature split thresholds along the traverse path, and each $l$ and $r$ is either $\pm\infty$ or equals to one $\eta^{i_k}$ along the traverse path. We use $B(\mathcal{C}(x)) = \bigcap_{i \in \mathcal{C}(x)} B^i = \bigcap_{i \in \mathcal{C}(x)} (l_1^i, r_1^i] \times \cdots \times \bigcap_{i \in \mathcal{C}(x)} (l_d^i, r_d^i]$ to denote the Cartesian product of the intersection of $K$ bounding boxes on the ensemble.

**Definition 2.** (*Valid Tuple*) $\mathbb{C} = \{\mathcal{C}(x) \mid \forall x \in \mathbb{R}^d\}$ denotes the set of all possible tuples that correspond to at least one point in the input space, and $\mathcal{C} = (i^{(1)}, \dots, i^{(K)})$ is a *valid* tuple *iff.* $\mathcal{C} \in \mathbb{C}$.

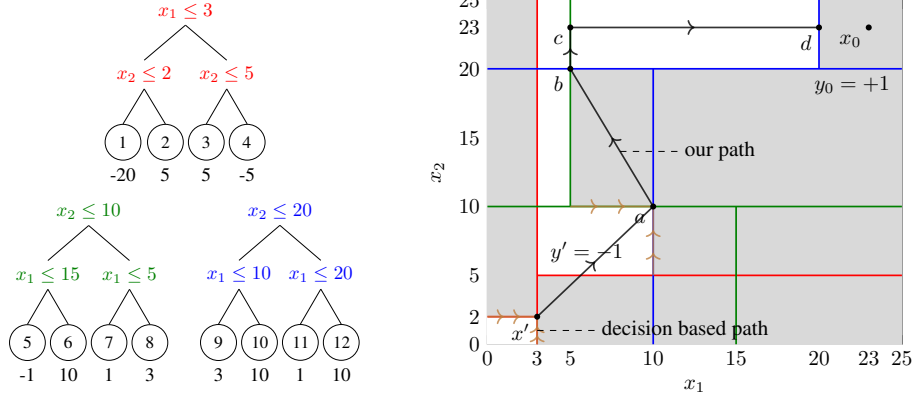

Figure 1: An ensemble defined on $[0, 25]^2$ and its corresponding decision boundaries on the input space. Numbers inside circles are indexes of leaves, and the number below each leaf is the corresponding prediction label $v^i$. For clarity we mark boundaries belong to tree 1, 2, 3 with red, green, and blue respectively, and fill $+1$ area with gray. $x_0$ is the victim example and $x'$ is an initial adversarial example. Assume we are optimizing $\ell_1$ perturbation, our method can reach $d$ by changing one leaf of the tuple at a time (black arrows). On the other hand, decision based attacks update the solution along the decision boundary, and easily fall into local minimums such as $x'$ and $a$ (brown arrows) since they look only at the continuous neighborhood. To move from $a$ to $b$, the path on decision boundary is $a \to (5, 10) \to b$, but since $a \to (5, 10)$ increases the distortion they won't find this path.

**Theorem 1.** *(Chen et al., 2019b)* The intersection $B(\mathcal{C})$ can also be written as the Cartesian product of $d$ intervals (similar to $B^i$), and $\mathcal{C} \in \mathbb{C} \iff B(\mathcal{C}) \neq \varnothing \iff \forall i, j \in \mathcal{C}, B^i \cap B^j \neq \varnothing$. (These concepts were used in Chen et al. 2019b for verification instead of attack.)

**Corollary 1.** (*Tuple to Example Distance*) The shortest distance between a valid leaf tuple and an example, defined as $\text{dist}_p(\mathcal{C}, x_0) = \min_{x \in B(\mathcal{C})} \|x - x_0\|_p$, can be solved in $O(d)$ time.

Observe $\mathcal{C}(x) = \mathcal{C}(x'), \ \forall x \in B(\mathcal{C}(x'))$ and we can transform the intractable number of $x$ into tractable number of leaf tuples $\mathcal{C}$. We abuse the notation $f(\mathcal{C})$ to denote the model prediction $\text{sign}(\sum_{i \in \mathcal{C}} v^i)$, which is a constant within $B(\mathcal{C})$. Combined with Corollary 1 we can rewrite Eq. (2) into the discrete form below. $\text{Neighbor}(\mathcal{C}')$ denotes the *neighborhood* space around $\mathcal{C}'$, which is a set of leaf tuples that *close* to $\mathcal{C}'$ in certain distance measurements. Fig. 1 presents an example to demonstrate that it's less likely to fall into local optimum on our newly defined neighborhood space.

$$\mathcal{C}'_{\text{new}} = \underset{\mathcal{C}}{\text{argmin}} \ \text{dist}_p(\mathcal{C}, x_0) \quad \text{s.t.} \quad \mathcal{C} \in \text{Neighbor}(\mathcal{C}') \cap \mathbb{C}, \ f(\mathcal{C}) \neq y_0. \tag{3}$$

### 3.2 Limitations of Naive Neighborhood Space Definitions

Now we discuss how to define $\text{Neighbor}(\mathcal{C}')$ to facilitate our attack. Intuitively the space should be efficient to compute, and has a reasonable coverage to avoid falling into local minimums too easily. In this section we discuss two naive approaches that fail on these two properties, and provide empirical results in Table 2.

**Enumerating all leaves is not efficient** (*NaiveLeaf*): Given current adversarial example $x'$ and its corresponding leaf tuple $\mathcal{C}' = (i^{(1)}, \ldots, i^{(K)})$, an intuitive approach is to change a single $i^{(t)}$ to a different leaf $i^{(t)}_{\text{new}}$. However the resulting tuple may not be valid, and we will have to query the ensemble to get a valid tuple in $\mathbb{C}$. We provide a possible implementation in Appendix D.2 where we move $x'$ to the closest point within $B^{i^{(t)}_{\text{new}}}$. NaiveLeaf requires multiple full model queries and takes $O(K \cdot 2^l \cdot Kl)$ time per iteration for $K$ trees of depth $l$, which is too time consuming (see Table 2).

**Mutating one feature at a time has poor coverage** (*NaiveFeature*): Given current adversarial example $x'$ and its corresponding leaf tuple $\mathcal{C}'$, another intuitive approach is to move $x'$ outside of $B(\mathcal{C}')$ on each feature dimension. This approach is efficient since there are at most $2d$ neighborhood, and each neighborhood is only different by one tree (assuming unique split thresholds). However the

Table 2: Average $\ell_2$ perturbation over 500 test examples on the standard (natural) GBDT models. ("*"): For a fair comparison we disabled the random noise optimization discussed in §3.5. Our LT-Attack searches in a subspace of NaiveLeaf so $\bar{r}_{\text{our}}$ is slightly larger, but it is significantly faster.

| Standard GBDT | NaiveLeaf | | NaiveFeature | | LT-Attack (Ours)* | | Ours vs. NaiveLeaf | |
|---|---|---|---|---|---|---|---|---|
| $\ell_2$ Perturbation | $\bar{r}_{\text{leaf}}$ | time | $\bar{r}$ | time | $\bar{r}_{\text{our}}$ | time | $\bar{r}_{\text{our}}/\bar{r}_{\text{leaf}}$ | Speedup |
| MNIST | .081 | 2.37s | .229 | .069s | .108 | .105s | 1.33 | 22.6X |
| F-MNIST | .080 | 3.93s | .181 | .061s | .096 | .224s | 1.20 | 17.5X |
| HIGGS | .008 | 3.17s | .011 | .023s | .009 | .031s | 1.13 | 102.3X |

method easily falls into local minimums due to the fact that each leaf is bounded by up to $l$ features jointly, thus it's unlikely to reach certain leaves by only changing one feature at a time. Taking Fig. 1 as an example and assume we are at $x'$, notice that $B(\mathcal{C}(x')) = [0,3] \times [0,2]$ and the algorithm stops here since both neighborhood $\{(3 + \epsilon, 2), (3, 2 + \epsilon)\}$ are not adversarial examples.

### 3.3 Define Neighborhood Space by Hamming Distance

In this section we introduce a neighborhood space through discrete hamming distance, and show that it's fast to compute and has good coverage. We define the distance $D(\mathcal{C}, \mathcal{C}')$ between two tuples as the number of different leaves, and the neighborhood of $\mathcal{C}'$ with distance $h$ by

$$\text{Neighbor}_h(\mathcal{C}') = \{\mathcal{C} \mid \forall \mathcal{C} \in \mathbb{C}, \ D(\mathcal{C}, \mathcal{C}') = h\}. \tag{4}$$

The intuition is that each tree can be queried independently, and we want to utilize such property by limiting the number of affected trees at each iteration. $\text{Neighbor}_h(\cdot)$ has a nice property where we can increase $h$ for larger search scope, or decrease $h$ to improve speed. Observe that $\text{Neighbor}_1(\mathcal{C}')$ is a subset of NaiveLeaf (minus invalid leaf tuples that requires an expensive model query), and a superset of NaiveFeature (plus leaf tuples that may affect multiple features). In experiments we are able to achieve good results with $\text{Neighbor}_1(\cdot)$, and we provide an empirical greedy algorithm in Appendix D.3 to estimate the minimal $h$ required to reach the exact solution.

### 3.4 An Efficient Algorithm for Solving Neighbor$_1(\cdot)$

We propose Leaf Tuple attack (LT-Attack) in Algorithm 1 that efficiently solves Eq. (3) through two additional concepts $T_{\text{Bound}}(\cdot)$ and $\text{Neighbor}_1^{(t)}(\cdot)$ as defined below. Let $\mathcal{C}'$ be any valid adversarial tuple, and assume unique feature split thresholds. By definition tuples in $\text{Neighbor}_1(\mathcal{C}')$ are only different from $\mathcal{C}'$ by one leaf, and we use $\text{Neighbor}_1^{(t)}(\mathcal{C}')$ to denote the neighborhood that has different prediction leaf on tree $t$. Formally

$$\text{Neighbor}_1^{(t)}(\mathcal{C}') = \{\mathcal{C} \mid \mathcal{C}^{(t)} \neq \mathcal{C}'^{(t)}, \ \mathcal{C} \in \text{Neighbor}_1(\mathcal{C}')\}. \tag{5}$$

**Definition 3.** (*Bound Trees*) Let $x' \in B(\mathcal{C}')$ be the example that minimizes $\text{dist}_p(\mathcal{C}', x_0)$, we denote the indexes of trees that *bounds* $x'$ by $T_{\text{Bound}}(\mathcal{C}') = \{t \mid \text{OnEdge}(x', B^{\mathcal{C}'^{(t)}}), \ \forall t \in \{1, \dots, K\}\}$. Here $\text{OnEdge}(x, B)$ is true *iff.* $x$ equals to the left or the right bound of $B$ on at least one dimension.

**Definition 4.** (*Advanced Neighborhood*) We denote the set of neighborhood with smaller (advanced) perturbation than $\mathcal{C}'$ by $\text{Neighbor}_1^+(\mathcal{C}') = \{\mathcal{C} \mid \mathcal{C} \in \text{Neighbor}_1(\mathcal{C}'), \ \text{dist}_p(\mathcal{C}, x_0) < \text{dist}_p(\mathcal{C}', x_0)\}$.

**Theorem 2.** (*Bound Neighborhood*) Let $\text{Neighbor}_{\text{Bound}}(\mathcal{C}') = \bigcup_{t \in T_{\text{Bound}}(\mathcal{C}')} \text{Neighbor}_1^{(t)}(\mathcal{C}')$, then $\text{Neighbor}_1^+(\mathcal{C}') \subseteq \text{Neighbor}_{\text{Bound}}(\mathcal{C}')$.

Theorem 2 suggests that we can solve Eq. (3) by searching over $\text{Neighbor}_{\text{Bound}}(\mathcal{C}')$ since it is a superset of the advanced neighborhood which leads to smaller perturbation. In general the algorithm consists of an outer loop and an inner $\text{Neighbor}_{\text{Bound}}(\cdot)$ function. The outer loop iterates until no better adversarial example can be found, while the inner function generates bound neighborhood with distance 1. The inner function computes $T_{\text{Bound}}$ and runs the top-down traverse for each $t \in T_{\text{Bound}}$ with the intersection of other $K - 1$ bounding boxes, denoted by $B^{(-t)}$. According to Theorem 1, a leaf node of $t$ is guaranteed to form a valid tuple if it has non-empty intersection with $B^{(-t)}$.

To efficiently obtain $B^{(-t)}$ we cache $K$ bounding boxes in $B'$, and for each feature dimension we maintain the sorted list of left and right bounds from $K$ boxes respectively. Note that $B^i$ of leaf node $i$

**Algorithm 1:** Our proposed LT-Attack for constructing adversarial examples.

**Data:** White-box model $f$, victim example $x_0, y_0$, initial adversarial example $x'$.

```
 1 begin
 2 │   r', C' ← dist_p(C(x'), x_0), C(x');
 3 │   B' ← BuildSortedBoxes(C', f);
   │     ▷ O(Kl log K) - B' maintains K sorted bounding
   │       boxes on d feature dimensions.
 4 │   has_better_neighbor ← True;
 5 │   while has_better_neighbor do
 6 │   │   N_1 ← Neighbor_Bound(C', B', f);
   │   │     ▷ See complexity in §3.4.1
 7 │   │   N'_1 ← {C | C ∈ N_1, f(C) ≠ y_0};
   │   │     ▷ O(|N_1|) - f(C) can be calculated from
   │   │       f(C') in O(1) using the diff leaf.
 8 │   │   r*, C* ← argmin_{r,C}{dist_p(C, x_0) | C ∈ N'_1};
   │   │     ▷ O(l|N'_1|) - dist_p(C, x_0) can be calculated
   │   │       from r' in O(l) using the diff leaf.
 9 │   │   has_better_neighbor ← r* < r';
10 │   │   if has_better_neighbor then
11 │   │   │   r', C' ← r*, C*;
12 │   │   │   B' ← B'.ReplaceBox(C*);
   │   │   │     ▷ O(l log K) - Remove and add one box.
13 │   │   end
14 │   end
15 │   return r', C'
16 end
```

```
17 Function Neighbor_Bound(C', B', f):
18 │   N_1 ← ∅;
19 │   T ← T_Bound(C', B');
   │     ▷ O(d log K) - Need O(log K) to get the first
   │       (tightest) tree on each dimension, and assume
   │       C' caches the closet x' to x_0. We give a
   │       closer complexity analysis for |T_Bound(·)| in
   │       the following section.
20 │   for t ∈ T do
21 │   │   B^(-t) ← B'.RemoveBox(t);
   │   │     ▷ O(l log K) - Remove the bounding box of
   │   │       tree t from the sorted list (lazily), the
   │   │       box has at most l non-infinite dimensions.
22 │   │   I ← {i | B^i ∩ B^(-t) ≠ ∅, i ∈ S^(t) \ C'^(t)};
   │   │     ▷ O(2^l) - Traverse tree t top-down and
   │   │       return leaves for Neighbor_1^(t)(C'). S^(t)
   │   │       denotes the set of leaves of tree t.
23 │   │   N_1 ← N_1 ∪ {C' | C'^(t) ← i, i ∈ I};
   │   │     ▷ O(|I|) - Construct the neighborhood tuple
   │   │       by replacing the t_th leaf. In practice,
   │   │       we only need to return the diff (t, i).
24 │   end
25 │   return N_1
26 end
```

comes from feature split thresholds along the top-down traverse path, thus it has at most $l$ non-infinite dimensions, where $l$ is the depth of the tree. In conclusion we can add/remove a bounding box $B^i$ to/from $B'$ in $O(l \log K)$ time. We provide time complexity for most operations in Algorithm 1 inline, and give a detailed analysis for the size of $\text{Neighbor}_{\text{Bound}}(C')$ in the next section. See Appendix D.1 for the algorithm generating random initial adversarial examples.

### 3.4.1 Size of the Bound Neighborhood

Our LT-Attack enumerates all leaf tuples in the bound neighborhood at each iteration, thus the complexity of each iteration largely depends on the size of $\text{Neighbor}_{\text{Bound}}(C')$. In this section we analyze the size $|\text{Neighbor}_{\text{Bound}}(C')|$ and show it will not be too large on real datasets.

**Corollary 2.** (*Size of Neighbor$_1^{(t)}(C')$*) Let $k^{(t)} = |\{\eta \in B_j^{(-t)}, (j, \eta) \in H(t)\}|$ be the number of feature split thresholds inside $B^{(-t)}$, we have $|\text{Neighbor}_1^{(t)}(C')| \leq 2^{\min(k^{(t)}, l)} - 1$. Here $B^{(-t)} = \bigcap_{i \in C', i \neq C'^{(t)}} B^i$ and $H(t)$ denotes the set of feature split thresholds on all internal nodes of tree $t$.

In practice, $k^{(t)} \ll l$ since $B^{(-t)}$ is the intersection of $K - 1$ bounding boxes and only covers a small region of the input space $\mathbb{R}^d$. $|T_{\text{Bound}}(C')| \leq d$ and is also usually small in real datasets, which can be explained by the intuition that some features are less *important* and could reach the same value as $x_0$ easily. Both $|T_{\text{Bound}}(C')|$ and $|\text{Neighbor}_1^{(t)}(C')|$ characterize the complexity of $|\text{Neighbor}_{\text{Bound}}(C')|$. We provide empirical statistics in Appendix A, which suggests that $|\text{Neighbor}_{\text{Bound}}(C')|$ has the similar complexity as a single full model query. For instance, on the MNIST dataset with 784 features and 400 trees we have mean $|\text{Neighbor}_{\text{Bound}}(C')| \approx 367.9$, and the algorithm stops in $\sim 159.4$ iterations when it cannot find a better neighborhood. As a comparison, decision based methods usually require hundreds of full model queries per iteration to estimate the update direction.

### 3.4.2 Convergence Guarantee with Neighbor$_1(\cdot)$

In this section $\bar{C}$ denotes the converged tuple when the outer loop stops, and we discuss the property of the converged solution. Trivially $\bar{C}$ has the minimal adversarial perturbation within $\text{Neighbor}_1(\bar{C})$, and we can show that the guarantee is actually stronger.

Table 3: Average $\ell_\infty$ and $\ell_2$ perturbation of 500 test examples (or the entire test set when its size is less than 500) on **standard (natural) GBDT models**. Datasets are ordered by training data size. **Bold** and <span style="color:blue">blue</span> highlight the best and the second best entries respectively (not including MILP). ("*"): Average of 50 examples due to long running time. ("⋆"): HSJA has fluctuating running time.

| Standard GBDT | SignOPT | | HSJA | | RBA-Appr | | Cube | | LT-Attack (Ours) | | MILP | | Ours vs. MILP | |
|---|---|---|---|---|---|---|---|---|---|---|---|---|---|---|
| $\ell_\infty$ Perturbation | $\bar{r}$ | time | $\bar{r}$ | time | $\bar{r}$ | time | $\bar{r}$ | time | $\bar{r}_{our}$ | time | $r^*$ | time | $\bar{r}_{our}/r^*$ | Speedup |
| breast-cancer | .258 | .308s | .256 | .070s | .247 | .0008s | .530 | .230s | **.235** | .001s | .222 | .013s | 1.06 | 13X |
| diabetes | .083 | .343s | .078 | .066s | .113 | .0009s | .080 | .240s | **.059** | .002s | .056 | .084s | 1.05 | 42X |
| MNIST2-6 | .480 | 2.73s | .277 | 1.23s | .963 | .155s | .143 | 2.43s | **.097** | .222s | .065 | 28.7s | 1.49 | 129.3X |
| ijcnn | .043 | .313s | .043 | .096s | .074 | .020s | .035 | .334s | **.033** | .007s | .031 | 6.60s | 1.06 | 942.9X |
| MNIST | .195 | 3.70s | .117 | 28.7s⋆ | .983 | 4.11s | .056 | 4.42s | **.029** | .237s | .014 | 375s* | 2.07 | 1582.3X |
| F-MNIST | .155 | 4.38s | .065 | 1.81s | .607 | 5.55s | .038 | 5.45s | **.028** | .370s | .013 | 15min* | 2.15 | 2473X |
| webspam | .013 | 1.01s | .023 | .445s | .051 | .720s | .003 | .866s | **.001** | .051s | .0008 | 27.5s | 1.25 | 539.2X |
| covtype | .047 | .508s | .074 | .209s | .086 | 3.05s | .036 | .958s | **.032** | .038s | .028 | 10min* | 1.14 | 15736.8X |
| HIGGS | .009 | .465s | .012 | .157s | .099 | 55.3s* | .005 | .862s | **.004** | .036s | .004 | 52min* | 1.00 | 87166.7X |

| Standard GBDT | SignOPT | | HSJA | | RBA-Appr | | Cube | | LT-Attack (Ours) | | MILP | | Ours vs. MILP | |
|---|---|---|---|---|---|---|---|---|---|---|---|---|---|---|
| $\ell_2$ Perturbation | $\bar{r}$ | time | $\bar{r}$ | time | $\bar{r}$ | time | $\bar{r}$ | time | $\bar{r}_{our}$ | time | $r^*$ | time | $\bar{r}_{our}/r^*$ | Speedup |
| breast-cancer | .310 | .811s | .370 | .072s | .352 | .0008s | .678 | .248s | **.283** | .001s | .280 | .011s | 1.01 | 11X |
| diabetes | .106 | .650s | .123 | .064s | .158 | .001s | .136 | .269s | **.077** | .003s | .073 | .055s | 1.05 | 18.3X |
| MNIST2-6 | 2.18 | 7.29s | 2.45 | 1.54s | 3.98 | .155s | .801 | 3.73s | **.245** | .235s | .183 | 2.52s | 1.34 | 10.7X |
| ijcnn | .051 | .544s | .052 | .094s | .112 | .020s | .067 | .355s | **.044** | .010s | .043 | 2.18s | 1.02 | 218X |
| MNIST | 1.20 | 8.71s | 1.45 | 23.7s⋆ | 5.38 | 4.11s | .310 | 6.67s | **.072** | .243s | .043 | 32.5s | 1.67 | 133.7X |
| F-MNIST | .870 | 9.57s | .581 | 2.03s | 3.85 | 5.48s | .225 | 9.20s | **.073** | .400s | .049 | 49.3s | 1.49 | 123.3X |
| webspam | .023 | 3.26s | .112 | .529s | .128 | .721s | .009 | 1.04s | **.002** | .053s | .002 | 3.17s | 1.00 | 59.8X |
| covtype | .061 | .976s | .123 | .217s | .129 | 3.03s | .070 | 1.06s | **.045** | .039s | .042 | 237s | 1.07 | 6076.9X |
| HIGGS | .015 | 1.02s | .015 | .154s | .196 | 55.5s* | .013 | .905s | **.008** | .037s | .007 | 13min* | 1.14 | 20621.6X |

**Theorem 3.** (*Convergence Guarantee*) Let $V^+ = \{i \mid i \in \mathcal{C},\ \mathcal{C} \in \mathrm{Neighbor}_1^+(\bar{\mathcal{C}})\}$ be the union of leaves appeared in the advanced neighborhood $\mathrm{Neighbor}_1^+(\bar{\mathcal{C}})$ (Definition 4), then

$$\bar{\mathcal{C}} \text{ is the optimum adversarial tuple within valid combinations of } V^+.$$

Note that $V^+$ is the union of leaves, and leaves from multiple tuples of distance 1 could form a new valid tuple of larger distance to $\bar{\mathcal{C}}$ (by combining the different leaves together). In other words Theorem 3 suggests that our solution is not only a local optimal in $\mathrm{Neighbor}_1(\bar{\mathcal{C}})$, but also better than certain tuples in $\mathrm{Neighbor}_h(\bar{\mathcal{C}})$ with $h > 1$. In our illustrated example Fig. 1, assume the algorithm converged at $\bar{\mathcal{C}} = \mathcal{C}(a) = (4, 5, 9)$ on $\ell_\infty$ norm, here $\mathrm{Neighbor}_1^+(\mathcal{C}(a)) = \{(4, 8, 9),\ (4, 5, 10)\}$. Theorem 3 claims that there is no better adversarial tuple from any valid combinations within $V^+ = \{4, 5, 8, 9, 10\}$ such as $(4, 8, 10)$, even though it is from $\mathrm{Neighbor}_2(\bar{\mathcal{C}})$.

### 3.5 Implementation Details

For the initial point, we draw 20 random initial adversarial examples from a Gaussian distribution, and optimize with a fine-grained binary search before feeding to the proposed algorithm. We return the best adversarial example found among them (see Appendix D.1 for details). The ensemble is likely to contain duplicate feature split thresholds even though it's defined on $\mathbb{R}^d$, for example it may come from the image space $[255]^d$ and scaled to $\mathbb{R}^d$. Duplicate split thresholds are problematic since we cannot move across the threshold without affecting multiple trees, and to overcome the issue we use a relaxed version of $\mathrm{Neighbor}_1(\cdot)$ to allow changing multiple trees at one iteration, as long as it's caused by the same split threshold. When searching for the best neighborhood it's likely to have perturbation ties in $\ell_\infty$ and $\ell_1$ norm, in this case we use a secondary $\ell_2$ norm to break the tie. Eq. (3) looks for the best tuple across all neighborhood which may be unnecessary at early stage of iterations. To improve the efficiency we sort feature dimensions by $\mathrm{abs}(x' - x_0)$ (large first), and terminate the search earlier if a better tuple was found in the top 1 feature. To escape converged local minimums we change each coordinate to a nearby value from Gaussian distribution with 0.1 probability, and continue the iteration if a better adversarial example was found within 300 trials.

## 4 Experimental Results

We evaluate the proposed algorithm on 9 public datasets (Smith et al., 1988; Lecun et al., 1998; Chang, Lin, 2011; Wang et al., 2012; Baldi et al., 2014; Xiao et al., 2017; Dua, Graff, 2017) with both the standard (natural) GBDT and RF models, and on an additional 10th dataset (Bosch, 2016) with

Table 4: Average $\ell_\infty$ and $\ell_2$ perturbation of 5000 test examples (or the entire test set when its size is less than 5000) on **robustly trained GBDT models**. Datasets are ordered by training data size. **Bold** and blue highlight the best and the second best entries respectively (not including MILP). ("*" / "⋆"): Average of 1000 / 500 examples due to long running time.

| Robust GBDT | SignOPT | | HSJA | | RBA-Appr | | Cube | | LT-Attack (Ours) | | MILP | | Ours vs. MILP | |
|---|---|---|---|---|---|---|---|---|---|---|---|---|---|---|
| $\ell_\infty$ Perturbation | $\bar{r}$ | time | $\bar{r}$ | time | $\bar{r}$ | time | $\bar{r}$ | time | $\bar{r}_{\text{our}}$ | time | $r^*$ | time | $\bar{r}_{\text{our}}/r^*$ | Speedup |
| breast-cancer | **.403** | .371s | .405 | .073s | .405 | **.002s** | .888 | .238s | .404 | .002s | .401 | .010s* | 1.01 | 5.6X |
| diabetes | .119 | .364s | .123 | .068s | .138 | **.001s** | .230 | .239s | **.113** | .003s | .112 | .039s* | 1.01 | 14.4X |
| MNIST2-6 | .588 | 3.06s | .470 | 1.30s | .671 | **.137s** | .337 | 2.15s | **.333** | .275s | .313 | 177s* | 1.06 | 641.6X |
| ijcnn | .032 | .353s | .030 | .105s | .032 | .018s | .027 | .313s | **.025** | .006s | .022 | 4.24s* | 1.14 | 759.6X |
| MNIST | .513 | 3.93s | .389 | 1.68s | .690 | 6.42s | .296 | 3.95s | **.290** | .234s | .270 | 20min* | 1.07 | 5067.5X |
| F-MNIST | .254 | 4.31s | .154 | 1.79s | .596 | 7.83s | .101 | 4.45s | **.095** | .412s | .076 | 74min* | 1.25 | 10778.5X |
| webspam | .047 | 1.00s | .043 | .414s | .061 | .641s | .020 | .756s | **.017** | .031s | .015 | 129s* | 1.13 | 4129.4X |
| covtype | .064 | .540s | .080 | .186s | .093 | 3.61s | .055 | .720s | **.047** | .047s | .045 | 14min* | 1.04 | 17164.9X |
| bosch | .343 | 3.28s | .337 | 1.42s | .533 | 1.22s | .158 | 2.49s | **.143** | .213s | .100 | 237s* | 1.43 | 1112X |
| HIGGS | .015 | .466s | .016 | .134s | .048 | 72.4s* | .012 | .644s | **.01** | .050s | .009 | 73min⋆ | 1.11 | 87149.2X |

| Robust GBDT | SignOPT | | HSJA | | RBA-Appr | | Cube | | LT-Attack (Ours) | | MILP | | Ours vs. MILP | |
|---|---|---|---|---|---|---|---|---|---|---|---|---|---|---|
| $\ell_2$ Perturbation | $\bar{r}$ | time | $\bar{r}$ | time | $\bar{r}$ | time | $\bar{r}$ | time | $\bar{r}_{\text{our}}$ | time | $r^*$ | time | $\bar{r}_{\text{our}}/r^*$ | Speedup |
| breast-cancer | .437 | .711s | .449 | .069s | .436 | **.002s** | .940 | .239s | **.434** | .002s | .431 | .011s* | 1.01 | 5.2X |
| diabetes | .142 | .591s | .150 | .061s | .161 | **.003s** | .274 | .240s | **.133** | .005s | .132 | .025s* | 1.01 | 4.8X |
| MNIST2-6 | 2.97 | 7.37s | 3.32 | 1.28s | 2.95 | **.156s** | 1.31 | 3.19s | **.971** | .438s | .762 | 25.0s* | 1.27 | 57.1X |
| ijcnn | .033 | .572s | .035 | .096s | .040 | .014s | .042 | .307s | **.030** | .006s | .025 | .853s* | 1.20 | 140.3X |
| MNIST | 3.08 | 9.14s | 3.04 | 1.61s | 4.07 | 5.11s | 1.33 | 6.26s | **.932** | .291s | .670 | 7min* | 1.39 | 1523.6X |
| F-MNIST | 1.67 | 9.27s | 1.34 | 1.64s | 3.72 | 7.01s | .500 | 7.01s | **.310** | .385s | .233 | 231s* | 1.33 | 600.8X |
| webspam | .097 | 3.24s | .100 | .431s | .148 | .589s | .068 | .869s | **.041** | .034s | .035 | 28.3s* | 1.17 | 840.6X |
| covtype | .076 | 1.11s | .104 | .196s | .137 | 3.26s | .096 | .726s | **.062** | .047s | .058 | 9min* | 1.07 | 11183.1X |
| bosch | .750 | 9.62s | 2.33 | 1.54s | 1.45 | 1.21s | .480 | 3.84s | **.258** | .232s | .214 | 28.0s* | 1.21 | 120.7X |
| HIGGS | .020 | .879s | .020 | .128s | .085 | 66.5s* | .023 | .580s | **.016** | .045s | .014 | 24min⋆ | 1.14 | 31715.5X |

the robustly trained GBDT. Datasets have a mix of small/large scale and binary/multi classification (statistics in Appendix A), and are normalized to $[0, 1]$ to make results comparable across datasets. We order datasets by training data size in all of our tables, where HIGGS is the largest dataset with 10.5 million training examples. All GBDTs were trained using the XGBoost framework (Chen, Guestrin, 2016) and we use the models provided by Chen et al. (2019b) as target models (except bosch). We compare with the following existing adversarial attacks that are applicable to tree ensembles:

• *SignOPT* (Cheng et al., 2020): The decision based attack that constructs adversarial examples based on hard-label black-box queries. We report the average distortion, denoted as $\bar{r}$ in the results (since the norm of adversarial example is an upper bound of minimal adversarial perturbation $r^*$).

• *HSJA* (Chen et al., 2019c): Another decision based attack for constructing adversarial examples.

• *RBA-Appr* (Yang et al., 2019): An approximate attack for tree ensembles that constructs adversarial examples by searching over training examples of the opposite class.

• *Cube* (Andriushchenko, Hein, 2019): An empirical attack for tree ensembles that constructs adversarial examples by stochastically changing a few coordinates to the $\ell_\infty$ boundary, and accepts the change if it decreases the functional margin. The method provides good experimental results in general, but lacks theoretical guarantee and could be unreliable on certain datasets such as breast-cancer. Cube doesn't support $\ell_2$ objective by default and we report the $\ell_2$ perturbation of the constructed adversarial examples from $\ell_\infty$ objective attacks.

• *LT-Attack (Ours)*: Our proposed attack that constructs adversarial examples for tree ensembles. We report the average distortion of the adversarial examples, denoted as $\bar{r}_{\text{our}}$ in the results.

• *MILP* (Kantchelian et al., 2015): The mixed-integer linear programming based method provides the exact minimal adversarial perturbation $r^*$ but could be very slow on large models.

We run our experiments with 20 threads per task. Conventionally black-box attacks measure efficiency by the number of queries, here we compare running time since it's difficult to quantify queries for white-box attacks. To minimize the efficiency variance between programming languages we feed an XGBoost model (Chen, Guestrin, 2016) to SignOPT, HSJA, and Cube, which has an efficient C++ implementation and supports multi-threading batch query. MILP uses a thin wrapper around the Gurobi Solver (Gurobi Optimization, 2020). Baseline methods spend majority of time on XGBoost model inference rather than Python code. For instance, on Fashion-MNIST, SignOPT, HSJA, Cube spent 72.8%, 57.3%, 73.4% of runtime in XGBoost library (C++) calls, respectively. HSJA and

Table 5: Average $\ell_2$ perturbation over 100 test examples on the **standard (natural) random forests (RF) models**. Datasets are ordered by training data size. **Bold** and blue highlight the best and the second best entries respectively (not including MILP).

| Standard RF | Cube | | LT-Attack (Ours) | | MILP | | Ours vs. MILP | |
|---|---|---|---|---|---|---|---|---|
| $\ell_2$ Perturbation | $\bar{r}$ | time | $\bar{r}_{\text{our}}$ | time | $r^*$ | time | $\bar{r}_{\text{our}}/r^*$ | Speedup |
| breast-cancer | 1.03 | .224s | **.413** | **.001s** | .402 | .008s | 1.03 | 8X |
| diabetes | .260 | .285s | **.151** | **.003s** | .146 | .042s | 1.03 | 14X |
| MNIST2-6 | .439 | 2.13s | **.207** | **.045s** | .194 | .071s | 1.07 | 1.6X |
| ijcnn | .046 | .336s | **.028** | **.003s** | .028 | .185s | 1.00 | 61.7X |
| MNIST | .057 | 2.88s | **.018** | **.057s** | .018 | 4.56s | 1.00 | 80X |
| F-MNIST | .141 | 3.51s | **.066** | **.080s** | .066 | 7.44s | 1.00 | 93X |
| webspam | .005 | .704s | **.003** | **.033s** | .003 | .664s | 1.00 | 20.1X |
| covtype | .087 | .700s | **.055** | **.040s** | .055 | 30.1s | 1.00 | 752.5X |
| HIGGS | .015 | .423s | **.009** | **.013s** | .009 | 6.66s | 1.00 | 512.3X |

SignOPT start with 1 initial adversarial example and run 100∼632 and 200 queries per iteration respectively to approximate the gradient, and Cube uses 20 initial examples which utilizes batch query. In Table 3 and Table 4 we show the empirical comparisons on $\ell_\infty$ and $\ell_2$, and provide $\ell_1$ results as well as the attack success rate in Appendix B due to the space limit. We can see that our method provides a tight upper bound $\bar{r}_{\text{our}}$ compared to the exact $r^*$ from MILP, which means that the adversarial examples found are very close to the one with minimal adversarial perturbation, and our method achieved 1,000∼80,000x speedup on some large models such as HIGGS. Our bound is tight especially in the $\ell_2$ case. For instance, on Fashion-MNIST our $\bar{r}/r^*$ ratio is 1.49x and 1.33x for standard and robustly trained models respectively, while the respective Cube ratio is 4.59x and 2.15x using ∼21x time, and the respective HSJA ratio is 11.86x and 5.75x using ∼4.5x time. For completeness we also include verification results (Chen et al., 2019b; Wang et al., 2020) in Appendix B since they are not directly comparable to adversarial attacks (they output lower bounds of the minimal adversarial perturbation while attacks aim to output an upper bound). To demonstrate the capability on general tree based ensembles we also conduct experiments on the standard (natural) random forests (RFs). We present the average $\ell_2$ perturbation in Table 5, where the $\bar{r}_{\text{our}}/r^*$ ratio is close to 1 across all datasets. Additional experimental results on $\ell_\infty$ perturbation can be found in Appendix B.1, and we include model parameters as well as statistics in Appendix A.

To study the impact of using different number of initial examples, we conduct the experiments with $\{1, 2, 4, 6, 10, 20, 40\}$ initial examples on SignOPT, HSJA, Cube, and LT-Attack, allocating 2 threads per task. We report the smallest (best) adversarial perturbation among those initial examples. Using more initial examples could lead to smaller (better) adversarial perturbation, but requires linearly increasing computational cost. Fig. 2 presents the $\ell_2$ perturbation vs. runtime per test example in log scale, where our method is able to construct small (good) adversarial examples (y-axis) with a few initial examples, and can be orders of magnitude faster than other methods in the meantime (x-axis).

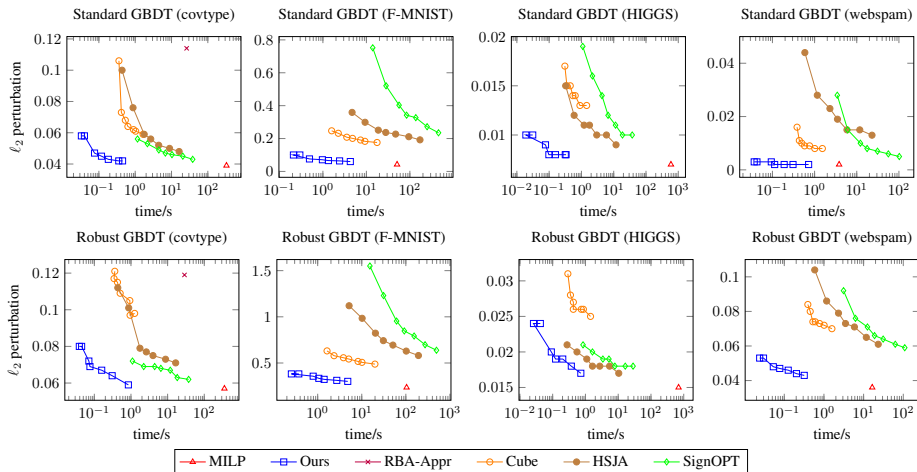

Figure 2: Average $\ell_2$ perturbation of 50 test examples vs. runtime per test example in log scale. Methods on the bottom-left corner are better.

## Broader Impact

To the best of our knowledge, this is the first practical attack algorithm (in terms of both computational time and solution quality) that can be used to evaluate the robustness of tree ensembles. The study of robustness training algorithms for tree ensemble models have been difficult due to the lack of attack tools to evaluate their robustness, and our method can serve as the benchmark tool for robustness evaluation (similar to FGSM, PGD and C&W attacks for neural networks) (Goodfellow et al., 2015; Madry et al., 2018; Carlini, Wagner, 2017) to stimulate the research in the robustness of tree ensembles.

## Acknowledgments and Disclosure of Funding

We acknowledge the support by NSF IIS-1901527, IIS-2008173, ARL-0011469453, Google Cloud and Facebook.

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
