[Supplementary Material]

# A  Dataset and Model Statistics

We use 9 datasets and pre-trained models provided in Chen et al. (2019b), which can be downloaded from `https://github.com/chenhongge/RobustTrees`. Table 6 summarized the statistics of the datasets as well as the standard (natural) GBDT models, and we report the average complexity statistics for $|\operatorname{Neighbor_{Bound}}(\cdot)|$ from 500 test examples. For multi-class datasets we count trees either belong to the victim class or the class of the initial adversarial example. Datasets may contain duplicate feature split thresholds and the extra complexity is covered in the statistics. We disabled the random noise optimization discussed in §3.5 to provide a cleaner picture of Algorithm 1. We train standard (natural) RF models using XGBoost's native RF APIs[1], and provide the statistics in Table 7.

Table 6: The average complexity statistics for $|\operatorname{Neighbor_{Bound}}(\cdot)|$ from 500 test examples.

| Dataset | features | classes | trees | depth $l$ | iterations | $|T_{Bound}(\cdot)|$ | $|\operatorname{Neighbor}_1^{(t)}(\cdot)|$ | $|\operatorname{Neighbor}_{Bound}(\cdot)|$ |
|---|---|---|---|---|---|---|---|---|
| breast-cancer | 10 | 2 | 4 | 6 | 2.1 | 3.2 | 5.2 | 9.2 |
| diabetes | 8 | 2 | 20 | 5 | 6.3 | 6.1 | 3.4 | 10.6 |
| MNIST2-6 | 784 | 2 | 1,000 | 4 | 121.7 | 374.2 | 14.9 | 256.5 |
| ijcnn | 22 | 2 | 60 | 8 | 26.5 | 7.4 | 3.3 | 17.8 |
| MNIST | 784 | 10 | 400 | 8 | 159.4 | 124.7 | 5.0 | 367.9 |
| F-MNIST | 784 | 10 | 400 | 8 | 236.8 | 149.1 | 6.5 | 717.4 |
| webspam | 254 | 2 | 100 | 8 | 100.7 | 37.0 | 3.8 | 129.7 |
| covtype | 54 | 7 | 160 | 8 | 36.7 | 30.8 | 10.6 | 39.2 |
| HIGGS | 28 | 2 | 300 | 8 | 107.1 | 13.5 | 2.1 | 24.0 |

Table 7: Parameters and statistics for datasets and the standard (natural) RFs.

| Dataset | train size | test size | trees | depth $l$ | subsampling | test acc. |
|---|---|---|---|---|---|---|
| breast-cancer | 546 | 137 | 4 | 6 | .8 | .974 |
| diabetes | 614 | 154 | 25 | 8 | .8 | .775 |
| MNIST2-6 | 11,876 | 1,990 | 1000 | 4 | .8 | .963 |
| ijcnn | 49,990 | 91,701 | 100 | 8 | .8 | .919 |
| MNIST | 60,000 | 10,000 | 400 | 8 | .8 | .907 |
| F-MNIST | 60,000 | 10,000 | 400 | 8 | .8 | .823 |
| webspam | 300,000 | 50,000 | 100 | 8 | .8 | .948 |
| covtype | 400,000 | 180,000 | 160 | 8 | .8 | .745 |
| HIGGS | 10,500,000 | 500,000 | 300 | 8 | .8 | .702 |

# B  Supplementary Experiments

## B.1  Additional Experimental Results on Random Forests

Table 8: Average $\ell_\infty$ perturbation over 100 test examples on the **standard (natural) random forests (RF) models**. Datasets are ordered by training data size. **Bold** and blue highlight the best and the second best entries respectively (not including MILP).

| Standard RF | Cube | | LT-Attack (Ours) | | MILP | | Ours vs. MILP | |
|---|---|---|---|---|---|---|---|---|
| $\ell_\infty$ Perturbation | $\bar{r}$ | time | $\bar{r}_{\mathrm{our}}$ | time | $r^*$ | time | $\bar{r}_{\mathrm{our}}/r^*$ | Speedup |
| breast-cancer | .797 | .208s | **.340** | **.001s** | .332 | .008s | 1.02 | 8X |
| diabetes | .159 | .271s | **.111** | **.002s** | .103 | .054s | 1.08 | 27X |
| MNIST2-6 | .135 | 1.85s | **.130** | **.041s** | .121 | .335s | 1.07 | 8.2X |
| ijcnn | .032 | .340s | **.026** | **.003s** | .026 | .338s | 1.00 | 112.7X |
| MNIST | .017 | 1.98s | **.010** | **.056s** | .009 | 21.4s | 1.11 | 382.1X |
| F-MNIST | **.036** | 2.57s | .036 | **.084s** | .032 | 34.1s | 1.13 | 406X |
| webspam | .004 | .652s | **.002** | **.023s** | .002 | 2.63s | 1.00 | 114.3X |
| covtype | .050 | .684s | **.048** | **.037s** | .048 | 72.2s | 1.00 | 1951.4X |
| HIGGS | .008 | .389s | **.007** | **.011s** | .006 | 20.9s | 1.17 | 1900X |

## B.2  Attack Success Rate

We present attack success rate in Fig. 3, which is calculated as the ratio of constructed adversarial examples that have smaller perturbation than the thresholds. We use 50 test examples for MILP due to long running time, and 500 test examples for other methods.

Figure 3: Attack success rate vs. perturbation thresholds.

## B.3  $\ell_\infty$ Perturbation Using Different Number of Initial Examples

Fig. 4 presents the average $\ell_\infty$ perturbation of 50 test examples vs. runtime per test example in log scale. We plot the results for SignOPT, HSJA, Cube, and LT-Attack on $\{1, 2, 4, 6, 10, 20, 40\}$ initial examples, using 2 threads per task. Initial examples are not applicable to RBA-Appr and MILP thus we only plot a single point for each method.

Figure 4: Average $\ell_\infty$ perturbation of 50 test examples vs. runtime per test example in log scale. Methods on the bottom-left corner are better.

## B.4  Experimental Results for $\ell_1$ Norm and Verification

Table 9 presents the experimental results on $\ell_1$ norm perturbation. Cube doesn't support $\ell_1$ objective by default and we report the $\ell_1$ perturbation of the constructed adversarial examples from $\ell_\infty$ objective attacks. For completeness we include verification results (Chen et al., 2019b; Wang et al., 2020) in Table 9 and Table 10, which output lower bounds of the minimal adversarial perturbation denoted as $\underline{r}$ (in contrast to adversarial attacks that aim to output an upper bound $\bar{r}$).

Table 9: Average $\ell_1$ perturbation over 50 test examples on the **standard (natural) GBDT models** and **robustly trained GBDT models**. Datasets are ordered by training data size. **Bold** and blue highlight the best and the second best entries respectively (not including MILP and Verification).

| Standard GBDT | SignOPT | | RBA-Appr | | Cube | | LT-Attack (Ours) | | MILP | | Verification | | Ours vs. MILP | |
|---|---|---|---|---|---|---|---|---|---|---|---|---|---|---|
| $\ell_1$ Perturbation | $\bar{r}$ | time | $\bar{r}$ | time | $\bar{r}$ | time | $\bar{r}_{\text{our}}$ | time | $r^*$ | time | $\underline{r}$ | time | $\bar{r}_{\text{our}}/r^*$ | Speedup |
| breast-cancer | .413 | .686s | .535 | **.0002s** | 1.02 | .234s | **.372** | .002s | .372 | .012s | .367 | .003s | 1.00 | 6.X |
| diabetes | .183 | 1.04s | .294 | **.0005s** | .290 | .238s | **.131** | .003s | .126 | .080s | .095 | .133s | 1.04 | 26.7X |
| MNIST2-6 | 35.4 | 6.81s | 25.9 | **.109s** | 3.73 | 3.47s | **.783** | .230s | .568 | 2.51s | .057 | 1.17s | 1.38 | 10.9X |
| ijcnn | .075 | .889s | .286 | .016s | .247 | .340s | **.064** | .008s | .060 | 3.28s | .044 | 1.68s | 1.07 | 410.X |
| MNIST | 22.8 | 8.77s | 46.5 | 3.13s | 2.16 | 7.11s | **.341** | .267s | .207 | 56.8s | .013 | 5.40s | 1.65 | 212.7X |
| F-MNIST | 13.8 | 9.55s | 42.7 | 4.57s | 1.20 | 9.04s | **.260** | **.479s** | .181 | 70.3s | .013 | 7.19s | 1.44 | 146.8X |
| webspam | .287 | 3.43s | .614 | .630s | .036 | 1.01s | **.006** | .062s | .004 | 4.17s | .0003 | 6.40s | 1.50 | 67.3X |
| covtype | .074 | 1.74s | .223 | 2.29s | .132 | 1.01s | **.057** | **.039s** | .052 | 314s | .024 | 2.26s | 1.10 | 8051.3X |
| HIGGS | .050 | 1.20s | .611 | 44.7s | .039 | .854s | **.016** | .049s | .013 | 25min | .002 | 7.36s | 1.23 | 30183.7X |

| Robust GBDT | SignOPT | | RBA-Appr | | Cube | | LT-Attack (Ours) | | MILP | | Verification | | Ours vs. MILP | |
|---|---|---|---|---|---|---|---|---|---|---|---|---|---|---|
| $\ell_1$ Perturbation | $\bar{r}$ | time | $\bar{r}$ | time | $\bar{r}$ | time | $\bar{r}_{\text{our}}$ | time | $r^*$ | time | $\underline{r}$ | time | $\bar{r}_{\text{our}}/r^*$ | Speedup |
| breast-cancer | .654 | .869s | .598 | **.0008s** | 1.23 | .210s | **.574** | .002s | .574 | .008s | .506 | .001s | 1.00 | 4.X |
| diabetes | .201 | .667s | .228 | **.001s** | .514 | .235s | **.189** | .002s | .189 | .028s | .166 | .007s | 1.00 | 14.X |
| MNIST2-6 | 23.8 | 6.51s | 17.8 | **.106s** | 6.69 | 3.47s | **2.76** | .523s | 1.78 | 23.2s | .381 | 2.91s | 1.55 | 44.4X |
| ijcnn | .076 | .693s | .205 | .015s | .233 | .337s | **.067** | **.007s** | .065 | 1.02s | .043 | .279s | 1.03 | 145.7X |
| MNIST | 57.4 | 7.93s | 32.7 | 3.70s | 9.76 | 8.35s | **4.00** | .455s | 1.72 | 13min | .270 | 8.61s | 2.33 | 1652.7X |
| F-MNIST | 28.6 | 10.4s | 41.3 | 4.96s | 3.70 | 9.96s | **1.20** | .477s | .720 | 244s | .077 | 13.6s | 1.67 | 511.5X |
| webspam | .186 | 3.65s | .540 | .522s | .309 | 1.00s | **.119** | .037s | .073 | 67.7s | .014 | 2.17s | 1.63 | 1829.7X |
| covtype | .097 | 1.65s | .217 | 2.43s | .241 | 1.02s | **.080** | **.075s** | .071 | 437s | .033 | 3.12s | 1.13 | 5826.7X |
| HIGGS | .033 | 1.12s | .226 | 43.6s | .071 | .839s | **.028** | **.052s** | .021 | 40min | .006 | 5.55s | 1.33 | 46576.9X |

Table 10: Average $\ell_\infty$ and $\ell_2$ perturbation over 500 test examples on the standard (natural) GBDT models and robustly trained GBDT models. ("*"): Average of 50 examples due to long running time.

| Standard GBDT | LT-Attack (Ours) | | MILP | | Verification | |
|---|---|---|---|---|---|---|
| $\ell_\infty$ Perturbation | $\bar{r}_{\text{our}}$ | time | $r^*$ | time | $\underline{r}$ | time |
| breast-cancer | .235 | .001s | .222 | .013s | .220 | .002s |
| diabetes | .059 | .002s | .056 | .084s | .047 | .910s |
| MNIST2-6 | .097 | .222s | .065 | 28.7s | .053 | 1.27s |
| ijcnn | .033 | .007s | .031 | 6.60s | .027 | 6.25s |
| MNIST | .029 | .237s | .014 | 375s* | .011 | 9.38s |
| F-MNIST | .028 | .370s | .013 | 15min* | .012 | 6.96s |
| webspam | .001 | .051s | .0008 | 27.5s | .0002 | 9.79s |
| covtype | .032 | .038s | .028 | 10min* | .021 | 4.21s |
| HIGGS | .004 | .036s | .004 | 52min* | .002 | 13.2s |

| Robust GBDT | LT-Attack (Ours) | | MILP | | Verification | |
|---|---|---|---|---|---|---|
| $\ell_\infty$ Perturbation | $\bar{r}_{\text{our}}$ | time | $r^*$ | time | $\underline{r}$ | time |
| breast-cancer | .415 | .001s | .415 | .008s | .414 | .001s |
| diabetes | .122 | .002s | .121 | .036s | .119 | .011s |
| MNIST2-6 | .331 | .302s | .317 | 98.7s | .311 | 29.4s |
| ijcnn | .038 | .006s | .036 | 3.60s | .032 | .799s |
| MNIST | .298 | .315s | .278 | 13min* | .255 | 7.47s |
| F-MNIST | .098 | .403s | .078 | 29min* | .075 | 13.3s |
| webspam | .016 | .038s | .014 | 51.2s | .011 | 5.85s |
| covtype | .047 | .053s | .044 | 518s* | .031 | 3.24s |
| HIGGS | .01 | .054s | .009 | 45min* | .005 | 8.42s |

| Standard GBDT | LT-Attack (Ours) | | MILP | | Verification | |
|---|---|---|---|---|---|---|
| $\ell_2$ Perturbation | $\bar{r}_{\text{our}}$ | time | $r^*$ | time | $\underline{r}$ | time |
| breast-cancer | .283 | .001s | .280 | .011s | .277 | .002s |
| diabetes | .077 | .003s | .073 | .055s | .058 | .458s |
| MNIST2-6 | .245 | .235s | .183 | 2.52s | .058 | 1.06s |
| ijcnn | .044 | .010s | .043 | 2.18s | .030 | 4.62s |
| MNIST | .072 | .243s | .043 | 32.5s | .013 | 6.81s |
| F-MNIST | .073 | .400s | .049 | 49.3s | .013 | 6.72s |
| webspam | .002 | .053s | .002 | 3.17s | .0002 | 8.95s |
| covtype | .045 | .039s | .042 | 237s | .023 | 2.96s |
| HIGGS | .008 | .037s | .007 | 13min* | .002 | 10.3s |

| Robust GBDT | LT-Attack (Ours) | | MILP | | Verification | |
|---|---|---|---|---|---|---|
| $\ell_2$ Perturbation | $\bar{r}_{\text{our}}$ | time | $r^*$ | time | $\underline{r}$ | time |
| breast-cancer | .452 | .001s | .452 | .007s | .450 | .001s |
| diabetes | .144 | .002s | .143 | .024s | .130 | .009s |
| MNIST2-6 | .968 | .401s | .803 | 17.3s | .358 | 4.42s |
| ijcnn | .048 | .007s | .046 | .728s | .035 | .575s |
| MNIST | .996 | .395s | .701 | 200s | .273 | 10.1s |
| F-MNIST | .326 | .468s | .251 | 99.3s | .079 | 13.5s |
| webspam | .039 | .036s | .033 | 12.0s | .012 | 4.72s |
| covtype | .063 | .054s | .059 | 280s | .033 | 3.10s |
| HIGGS | .016 | .054s | .015 | 15min* | .006 | 7.16s |

## C Proofs

### C.1 Proof of Theorem 2

*Proof.* By contradiction. Given adversarial tuple $\mathcal{C}'$ and victim example $x_0$, assume

$$\exists \mathcal{C}_1 \in \text{Neighbor}_1^+(\mathcal{C}') \quad \text{s.t.} \quad \mathcal{C}_1 \notin \text{Neighbor}_{\text{Bound}}(\mathcal{C}').$$

Assume $p \in \{1, 2, \infty\}$. Let $x_1 = \operatorname{argmin}_{x \in B(\mathcal{C}_1)} \|x - x_0\|_p$, $x' = \operatorname{argmin}_{x \in B(\mathcal{C}')} \|x - x_0\|_p$, and let $J$ be the set of dimensions that $x_1$ is closer to $x_0$:

$$J = \{j \mid |x_{1,j} - x_{0,j}| < |x'_j - x_{0,j}|\}.$$

According to the definition of $\text{Neighbor}_1^+(\mathcal{C}')$ we have $\text{dist}_p(\mathcal{C}_1, x_0) < \text{dist}_p(\mathcal{C}', x_0)$, and consequently $J \neq \varnothing$. We choose any $j' \in J$ and for cleanness we use $(l', r')$ to denote the interval from $B(\mathcal{C}')$ on $j'_{\text{th}}$ dimension, and let $d_0 = x_{0,j'}, d_1 = x_{1,j'}, d' = x'_{j'}$.

Observe $d_0 \notin (l', r']$, otherwise we have $|d' - d_0| = 0$ and $|d_1 - d_0|$ cannot be smaller. W.l.o.g. assume $d_0$ is on the right side of the interval, i.e., $r' < d_0$, then according to the $\mathrm{argmin}$ property of $x'$ we have $d' = r'$.

Recall that $B(\mathcal{C}')$ is the intersection of $K$ bounding boxes from the ensemble, then

$$\exists t' \in [K] \quad \text{s.t.} \quad B_{j'}^{\mathcal{C}'^{(t)}}.r = r'.$$

Observe $r' = d' < d_1$ since $d_0$ is on the right side of $d'$ and $d_1$ has smaller distance to $d_0$, which means $\mathcal{C}_1$ has different leaf than $\mathcal{C}'$ on tree $t$. Also according to the above equation $t' \in T_{\mathrm{Bound}}(\mathcal{C}')$, and $\mathcal{C}_1 \in \mathrm{Neighbor}_1^{(t')}(\mathcal{C}')$, thus $\mathcal{C}_1 \in \bigcup_{t \in T_{\mathrm{Bound}}(\mathcal{C}')} \mathrm{Neighbor}_1^{(t)}(\mathcal{C}') = \mathrm{Neighbor}_{\mathrm{Bound}}(\mathcal{C}')$, contradiction.

$\square$

## C.2 Proof of Corollary 2

*Proof.* According to Theorem 1 $B^{(-t)}$ is the intersection of $K - 1$ bounding boxes and can be written as the Cartesian product of $d$ intervals, we call it a *box*. Each tree $t$ of depth $l$ splits the $\mathbb{R}^d$ space into up to $2^l$ non-overlapping axis-aligned boxes, and $|\mathrm{Neighbor}_1^{(t)}(\mathcal{C}')| + 1$ (plus current box) corresponds to the number of boxes that has non-empty intersection with $B^{(-t)}$, thus $|\mathrm{Neighbor}_1^{(t)}(\mathcal{C}')| \leq 2^l - 1$.

Observe that $k^{(t)}$ axis-aligned feature split thresholds can split $\mathbb{R}^d$ into at most $2^{k^{(t)}}$ non-overlapping boxes, assuming $d \geq k^{(t)}$, and the maximum can be reached by having at most 1 split threshold on each dimension. In conclusion there are at most $2^{\min(k^{(t)}, l)}$ boxes that has non-empty intersection with $B^{(-t)}$, thus $|\mathrm{Neighbor}_1^{(t)}(\mathcal{C}')| \leq 2^{\min(k^{(t)}, l)} - 1$ (minus the current box). $\square$

## C.3 Proof of Theorem 3

*Proof.* By contradiction. Let $x_0, y_0$ be the victim example and assume

$$\exists \mathcal{C}^* \in (V^+)^K \cap \mathbb{C} \quad \text{s.t.} \quad f(\mathcal{C}^*) \neq y_0 \wedge \mathrm{dist}_p(\mathcal{C}^*, x_0) < \mathrm{dist}_p(\mathcal{C}', x_0).$$

Assume $p \in \{1, 2, \infty\}$. Recall $f(\mathcal{C}) = sign(\sum_{i \in \mathcal{C}} v^i)$, we compute the tree-wise prediction difference

$$v_{\mathrm{diff}}^{(t)} = v^{\mathcal{C}^{*(t)}} - v^{\mathcal{C}'^{(t)}}, \ t \in [K].$$

Let $t_{\min}$ be the tree with the smallest functional margin difference

$$t_{\min} = \operatorname*{argmin}_{t \in [K]} y_0 \cdot v_{\mathrm{diff}}^{(t)}.$$

We construct a tuple $\mathcal{C}_1$ which is the same as $\mathcal{C}'$ except on $t_{\min}$, where $\mathcal{C}_1^{(t_{\min})} = \mathcal{C}^{*(t_{\min})}$, consequently we have

$$\mathcal{C}_1 \in \mathbb{C} \wedge \mathrm{dist}_p(\mathcal{C}_1, x_0) < \mathrm{dist}_p(\mathcal{C}', x_0).$$

Now we show $f(\mathcal{C}_1) \neq y_0$, or $y_0 \sum_{i \in \mathcal{C}_1} v^i < 0$:

  i. Case $y_0 \cdot v_{\mathrm{diff}}^{(t_{\min})} \leq 0$. Then

  $$y_0 \sum_{i \in \mathcal{C}_1} v^i = y_0 \sum_{i \in \mathcal{C}'} v^i + y_0 \cdot v_{\mathrm{diff}}^{(t_{\min})} \leq y_0 \sum_{i \in \mathcal{C}'} v^i < 0.$$

  ii. Case $y_0 \cdot v_{\mathrm{diff}}^{(t_{\min})} > 0$. Then

  $$y_0 \sum_{i \in \mathcal{C}_1} v^i = y_0 \sum_{i \in \mathcal{C}'} v^i + y_0 \cdot v_{\mathrm{diff}}^{(t_{\min})} < y_0 \sum_{i \in \mathcal{C}'} v^i + K y_0 \cdot v_{\mathrm{diff}}^{(t_{\min})}$$

  $$\leq y_0 \sum_{i \in \mathcal{C}'} v^i + y_0 \sum_{t \in [K]} v_{\mathrm{diff}}^{(t)} = y_0 \sum_{i \in \mathcal{C}^*} v^i < 0.$$

In conclusion $\mathcal{C}_1$ is a valid adversarial tuple within $\mathrm{Neighbor}_1(\mathcal{C}')$ and has smaller perturbation than $\mathcal{C}'$, thus the algorithm won't stop. $\square$

# D Supplementary Algorithms

## D.1 Generating Initial Adversarial Examples for LT-Attack

---

**Algorithm 2:** Generating Initial Adversarial Examples for LT-Attack

---

**Data:** Target white-box model $f$, victim example $x_0$.

1 **begin**
2      $y_0 \leftarrow f(x_0)$;
3      $r'$, $\mathcal{C}' \leftarrow MAX$, *None*;
4      *num_attack* $\leftarrow 20$;
5      **for** $i \leftarrow 1, \ldots, num\_attack$ **do**
6          **do**
7              $x' \leftarrow x_0 + Normal(0,1)^d$;
8          **while** $f(x') = y_0$;
9          $x' \leftarrow BinarySearch(x', x_0, f)$;
         ▷ Do a fine-grained binary search between $x_0$ and $x'$ to optimize the initial perturbation of $x'$. Similar to $g(\theta)$ proposed by Cheng et al. (2019).
10          $r^*$, $\mathcal{C}^* \leftarrow LT\text{-}Attack(f, x_0, x')$;
11          **if** $r^* < r'$ **then**
12              $r'$, $\mathcal{C}' \leftarrow r^*$, $\mathcal{C}^*$;
13          **end**
14      **end**
15      **return** $r'$, $\mathcal{C}'$
16 **end**

---

## D.2 Algorithm for NaiveLeaf

---

**Algorithm 3:** Compute NaiveLeaf

---

**Data:** Target white-box model $f$, current adversarial example $x'$, victim example $x_0$.
**Result:** The NaiveLeaf neighborhood of $\mathcal{C}(x')$

1 **begin**
2      $(i^{(1)}, \ldots, i^{(K)}) \leftarrow \mathcal{C}(x')$;
3      $N \leftarrow \varnothing$;
4      **for** $t \leftarrow 1 \ldots K$ **do**
5          **for** $i \in S^{(t)}, i \neq i^{(t)}$ **do**
             ▷ $S^{(t)}$ denotes the leaves of tree $t$.
6              $x_{\text{new}} \leftarrow x'$;
7              **for** $j, (l, r] \in B^i$ **do**
                 ▷ The $j_{\text{th}}$ dimension of the bounding box $B^i$, can be acquired from $f$.
8                  **if** $x_{new,j} \notin (l, r]$ **then**
9                      $x_{\text{new},j} \leftarrow \min(r, \max(l + \epsilon, x_{0,j}))$;
10                  **end**
11              **end**
12              $N \leftarrow N \cup \{\mathcal{C}(x_{\text{new}})\}$;
13          **end**
14      **end**
15      **return** $N$
16 **end**

---

## D.3 A Greedy Algorithm to Estimate the Minimum Neighborhood Distance

To understand the quality of constructed adversarial examples we use an empirical greedy algorithm to estimate the minimum *neighborhood distance* $h$ such that $\text{Neighbor}_h(\cdot)$ can reach the *exact* solution. Assume our method converged at $\mathcal{C}'$ and the optimum solution is $\mathcal{C}^*$, let $T_{\text{diff}} = \{t \mid \mathcal{C}'^{(t)} \neq \mathcal{C}^{*(t)}\}$ be the set of trees with different prediction, then the Hamming distance $\bar{h} = |T_{\text{diff}}|$ is a trivial upper bound where $\text{Neighbor}_{\bar{h}}(\mathcal{C}')$ can reach $\mathcal{C}^*$ with a single addition iteration. To estimate a realistic $h^\sim$ we want to find the disjoint split $T_{\text{diff}} = \cup_{i \in [k]} T_i$ such that we can mutate $\mathcal{C}'$ into $\mathcal{C}^*$ by changing trees in $T_i$ to match $\mathcal{C}^*$ at $i_{\text{th}}$ iteration. We make sure all intermediate tuples are valid and has strictly

decreasing perturbation as required by Eq. (3), and report $h^{\sim} = \max_{i \in [k]} |T_i|$. $h^{\sim}$ is an estimation of the minimum $h$ because we cannot guarantee the $\mathrm{argmin}$ constrain due to the large complexity. As shown in Table 11 we have $median(\bar{h}) = 23$ and $median(h^{\sim}) = 8$ on ensemble with 300 trees (HIGGS), which suggests that our method is *likely* to reach the exact optimum on half of the test examples through $\sim$3 additional iterations on $\mathrm{Neighbor}_8(\cdot)$. In this experiment we disabled the random noise optimization discussed in §3.5 to provide a cleaner picture of Algorithm 1.

---

**Algorithm 4:** A greedy algorithm to estimate the minimum neighborhood distance $h^{\sim}$.

**Data:** The model $f$, our adversarial point $x_{\mathrm{our}}$, exact MILP solution $x^*$.
**Result:** An estimation of neighborhood distance $h^{\sim}$.

1 **begin**
2    $\mathcal{C}_{\mathrm{our}}, \mathcal{C}^* \leftarrow \mathcal{C}(x_{\mathrm{our}}), \mathcal{C}(x^*)$;
3    $r^* \leftarrow \mathrm{dist}_p(\mathcal{C}^*, x_0)$;
4    $y^* \leftarrow f(x^*)$;
5    $h_{\min} \leftarrow D(\mathcal{C}_{\mathrm{our}}, \mathcal{C}^*)$ ;                   ▷ Hamming distance is the upper bound.
6    $I_{\mathrm{diff}} \leftarrow \{(v^{\mathcal{C}^{*(t)}} - v^{\mathcal{C}_{\mathrm{our}}^{(t)}}, \mathcal{C}_{\mathrm{our}}^{(t)}, \mathcal{C}^{*(t)}) \mid \mathcal{C}_{\mathrm{our}}^{(t)} \neq \mathcal{C}^{*(t)}, t \in [K]\}$;
      ▷ $I_{\mathrm{diff}}$ is the list of tuples in the form of (label diff, our leaf, MILP leaf).
7    $num\_trial \leftarrow 200$;
8    **for** $i \leftarrow 1, \ldots, num\_trial$ **do**
9      $h \leftarrow 0$;
10     $I \leftarrow shuffle(I_{\mathrm{diff}})$;
11     $\mathcal{C}_{\mathrm{tmp}} \leftarrow \mathcal{C}_{\mathrm{our}}$;
12     **while** $I \neq \varnothing$ **do**
13       $r_{last} \leftarrow \mathrm{dist}_p(\mathcal{C}_{\mathrm{tmp}}, x_0)$;
14       $\mathcal{C}_{\mathrm{tmp}} \leftarrow$ *pop the first tuple from $I$ with positive label diff and replace with MILP leaf*;
15       $h \leftarrow h + 1$;
16       **while** $\mathcal{C}_{tmp} \notin \mathbb{C}$ *or* $dist_p(\mathcal{C}_{tmp}, x_0)_p \notin [r^*, r_{last}]$ *or* $f(\mathcal{C}_{tmp}) \neq y^*$ **do**
         ▷ Making sure $\mathcal{C}_{\mathrm{tmp}}$ satisfies Equation 3 except the $\mathrm{argmin}$. We cannot guarantee
          $\mathrm{argmin}$ due to the high complexity. The *while* loop is guaranteed to exit since we
          can pop all tuples in $I$ to become $\mathcal{C}^*$.
17         $\mathcal{C}_{\mathrm{tmp}} \leftarrow$ *pop the first tuple from $I$ and replace with MILP leaf*;
18         $h \leftarrow h + 1$;
19       **end**
20     **end**
21     $h_{\min} \leftarrow \min(h_{\min}, h)$;
22    **end**
23    **return** $h_{min}$
24 **end**

---

Table 11: Convergence statistics for the standard (natural) GBDT models between our solution and the optimum MILP solution. We collect the data after the fine-grained binary search but before applying LT-Attack (Initial), and the data after LT-Attack (Converged). We disabled the random noise optimization discussed in §3.5.

| Dataset | Model | HammingDist $\bar{h}$ | | | | NeighborDist $h^{\sim}$ | | | |
|---|---|---|---|---|---|---|---|---|---|
| | | Initial | | Converged | | Initial | | Converged | |
| | # of trees | max | median | max | median | max | median | max | median |
| breast-cancer | 4 | 2 | 0 | 0 | 0 | 2 | 0 | 0 | 0 |
| diabetes | 20 | 10 | 3 | 6 | 0 | 5 | 1 | 4 | 0 |
| MNIST2-6 | 1000 | 676 | 490 | 347 | 172 | 172 | 46 | 64 | 36 |
| ijcnn | 60 | 27 | 10 | 16 | 3 | 18 | 2 | 8 | 2 |
| MNIST | 400 | 266 | 115 | 96 | 33 | 173 | 8 | 12 | 7 |
| F-MNIST | 400 | 237 | 174 | 100 | 56 | 82 | 12 | 18 | 10 |
| webspam | 100 | 88 | 56 | 36 | 16 | 75 | 7 | 9 | 4 |
| covtype | 160 | 84 | 23 | 67 | 9 | 82 | 7 | 65 | 6 |
| HIGGS | 300 | 190 | 62 | 125 | 23 | 181 | 12 | 121 | 8 |

## Footnotes

[1] `https://xgboost.readthedocs.io/en/release_1.0.0/tutorials/rf.html`