[Reviews · NeurIPS 2020]

Review 1

Summary and Contributions: The authors study the problem of efficient adversarial attack on tree based ensembles such as gradient boosting decision trees. They propose a greedy algorithm to iteratively optimize the adversarial example.

Strengths: The authors provide theoretical guarantees for their work. They compare to baseline methods in their empirical validation.

Weaknesses: The overall writing of the paper can greatly be improved as some parts of the paper is not clear or easy to understand. The experiments are uses a small test sample size so its difficult to judge how efficient the method will be on large data.

Correctness: The authors claim their method is practical and efficient in evaluating robustness on tree ensembles but report results on small data sample size. They are yet to establish how their method will work for larger datasets.

Clarity: No, the paper is not well written and is difficult to follow.

Relation to Prior Work: The authors do compare to prior works but it is not clear how their work differ from some of the prior works. The authors didn't do a good job of explaining the differences.

Reproducibility: Yes

Additional Feedback: Some suggestions to improve the overall organization of the paper and aid understanding: Re-organize the background and related work section to clearly state how your method differs from the different related works. The limitation/similarity of each method to yours. The motivation for the work is not well established or should be better explained. Table 1 and 2 should highlight best performing methods to make it easier to read. Additionally, Figure 2 is too compact and adequate explanation of the performance of the methods in the figure should be provided. ========================================= I acknowledge that I read the rebuttal provided by the authors.


Review 2

Summary and Contributions: This paper transforms the attack problem into a discrete search problem specially designed for tree ensembles, where the goal is to find a valid leaf tuple that leads to mis-classification while having the shortest distance to the original input. Experiments also prove the effectiveness of the method.

Strengths: 1. This paper transforms the attack problem into a discrete search problem specially designed for tree ensembles. 2. The method is faster than many current adversarial attack methods. 3. The method can provide smaller (better) adversarial examples than decision-based black-box attacks on general Lp (p = 1, 2, 1) norm perturbations.

Weaknesses: 1. It seems that this method is only suitable for white box attacks. When the attacked GBDT tree based model is unknown (including the depth and number of trees), this method is obviously not applicable. 2. The paper does not give the experiments on more challenging data sets to verify the method performance on the tree with higher complexity. 3. The comparison methods are based on Python time statistics, while your method is based on C + +, which does not seem to have fair comparison. 4. Lack of ablation experiments and visual samples, which makes the manuscript difficult to understand.

Correctness: The expression of **bounding boxes B(C)** in method is not very clear.

Clarity: There are no obvious grammatical and spelling problems.

Relation to Prior Work: It is clearly discussed the difference from the previous contributions.

Reproducibility: No

Additional Feedback: Can you explain why the points x’ and a in Figure 1 are local minimums? The point b is not in the neighbor of a? Why the searching of adversarial sample will stop in x’ and a?


Review 3

Summary and Contributions: The paper presents an efficient attack algorithm against tree ensembles using a greedy search algorithm. The search cost is cut down significantly by transforming the continuous input space into discrete leaf tuples. Empirical results demonstrate impressive speedup with a tighter perturbation bound compared to the existing attack algorithms.

Strengths: The proposed attack technique works both efficiently and effectively in searching for adversarial samples with minimum perturbation. Extensive empirical studies demonstrate the superiority of the proposed attack technique in comparison to the existing ones.

Weaknesses: The practical significance of minimizing perturbation needs to be better motivated.

Correctness: The claims are strongly supported by the experimental results.

Clarity: Yes, although there's room for improvement. For example, the abuse of symbol \mathcal{C} in the mix with an input point x in "Tuple to Example Distance" is quite disturbing. On page 1, "require large amount of queries" ---> a large number of

Relation to Prior Work: Yes.

Reproducibility: Yes

Additional Feedback: The conventional wisdom of crafting an adversarial sample x' for x is to ensure x' and x are as close as possible according to a predefined distance measure. In practice, unless x' is distorted so much that it becomes an outlier and may be easily detected, how much does an adversary really care about the amount of distortion unless it is tied directly to the cost? It seems more reasonable to constraint adversarial attacks on their ability to transfer to other learning models or their preservation of malicious utilities.


Review 4

Summary and Contributions: The paper proposes a more efficient method of generating adversarial examples (in the white-box setting) for tree ensembles (random forest, boosted trees, etc.) by exploiting the structure of the ensemble. After selecting a random adversarial example from a pool of initial candidates, their algorithm greedily updates the initially chosen adversarial example until it is "close" to the target input sample (in terms of l_p norm). They do this by performing an iterative search over the transformed input space of valid leaf tuples, and stopping when no more adversarial examples with a smaller l_p norm exist. The paper contributes a new adversarial attack, specifically designed for tree ensembles, that is more efficient than alternative methods while generating adversarial examples similar to exact methods.

Strengths: Tree ensembles are widely used and important models in many domains; this is evident by the consistent publication of new implementations of gradient boosted frameworks such as XGBoost, LightGBM, and CatBoost. The problem of generating quality adversarial examples is important and can lead to more robust models, and thus would be of particular relevance to this community. This work introduces a method which generates adversarial examples close to the exact method while being significantly faster than previous methods, sometimes by several orders of magnitude. The idea of discretizing the input space into a set of valid leaf tuples from which an iterative greedy search is performed appears novel in contrast to previous works. The approach has a solid theoretical grounding and performs a thorough empirical evaluation that includes comparisons to relevant alternative methods.

Weaknesses: The biggest variable to this approach seems to be the size of the neighborhood of possible adversarial examples when updating, given the current adversarial example. The choice of this variable introduces a tradeoff between efficiency and optimality. However, the authors do provide a greedy algorithm for estimating the neighborhood size to obtain optimal results, and they empirically show that a hamming distance of 1 between neighbors generally works well. Since this approach utilizes the structure of the ensemble to create adversarial examples, is it robust to changes in the tree structure? For example, a tree may have multiple possible structures that are functionally equivalent; thus, is this method robust to structural changes and provide adversarial examples that ultimately improve robustness of a tree ensemble trained on a given dataset? In the same vein, have the authors performed any experiments applying their approach to building robust tree ensembles? If so, how do the resulting models compare with other "robustly" trained tree ensembles trained on datasets augmented with adversarial examples generated by alternative methods?

Correctness: The claims appear to be correct and the empirical methodology correct. However, I would like to know if the authors ran experiments on more of the datasets for the random forest experiments, as less than half of the overall number of datasets were used for those experiments.

Clarity: The paper is well written and organized with a clear motivation and problem definition. The arguments of their methodology also follows a logical and intuitive progression.

Relation to Prior Work: The authors do a good job of contrasting their work with alternative methods; one of the biggest differences of this approach is the significant decrease in number of full model queries in comparison to other methods.

Reproducibility: Yes

Additional Feedback: The text in all of the tables and figures are a bit small and hard to read. It can also be a bit difficult to compare each method's metric to one another in each table; perhaps one table for utility and another for efficiency may increase legibility and comprehension. I have read the author responses and they have adequately addressed my concerns, and I will maintain my recommendation of "accept".

[Author Response · NeurIPS 2020]

We thank all reviewers for their valuable comments and suggestions. Here we focus on clarifying major concerns, and
will address all minor points (fix notations, typos, and improve legibility for tables and figures) in our next revision.

**[R1] 1) Larger sample size**: In Table A, we repeat our experiments on 5000 test examples for each dataset (or the
entire test set when its size is less than 5000), 10X larger than originally reported. We highlighted the best and the
second best methods. The average $\bar{r}$ are similar to Table 3 across all datasets, showing the effectiveness of our algorithm.
We had to use a different machine for this larger experiment so time is not comparable, but the speedups are also similar
to those in Table 3. We have 2 large datasets, HIGGS and Bosch (see reply to **[R3]**-1)). **2) Difference with prior**
**works**: Our major novelty is to discretize the input space into a set of valid leaf tuples, on which we perform the greedy
search. Table B highlights our differences. **3) Motivation:** We provide a strong attack as a tool for evaluating the
robustness of tree based models. (see reply to **[R4]**-1)). **4) Figure 2 explanation:** We run each method with different
number of random initial examples (x-axis). More initial examples lead to better attacks (smaller perturbation size on
y-axis), but runtime cost is higher. Methods on bottom-left corner are better. We will enlarge figures and explain more.

Table A: Average $\ell_\infty$ and $\ell_2$ perturbation of **5000** test examples on robustly trained GBDT models. **Bold** and blue highlight the best and the second best entries respectively (not including MILP). ("*" / "⋆"): Average of 1000 / 500 examples due to long running time.

| Robust GBDT | SignOPT | | HSJA | | RBA-Appr | | Cube | | LT-Attack (Ours) | | MILP | | Ours vs. MILP | |
|---|---|---|---|---|---|---|---|---|---|---|---|---|---|---|
| $\ell_\infty$ Perturbation | $\bar{r}$ | time | $\bar{r}$ | time | $\bar{r}$ | time | $\bar{r}$ | time | $\bar{r}_{our}$ | time | $r^*$ | time | $\bar{r}_{our}/r^*$ | Speedup |
| MNIST2-6 | .588 | 3.06s | .470 | 1.30s | .671 | **.137s** | .337 | 2.15s | **.333** | .275s | .313 | 177s* | 1.06 | 641.6X |
| breast-cancer | **.403** | .371s | .405 | .073s | .405 | **.002s** | .888 | .238s | .404 | .002s | .401 | .010s* | 1.01 | 5.6X |
| covtype | .064 | .540s | .080 | .186s | .093 | 3.61s | .055 | .720s | **.047** | **.047s** | .045 | 14min* | 1.04 | 17164.9X |
| diabetes | .119 | .364s | .123 | .068s | .138 | **.001s** | .230 | .239s | **.113** | .003s | .112 | .039s* | 1.01 | 14.4X |
| FMNIST | .254 | 4.31s | .154 | 1.79s | .596 | 7.83s | .101 | 4.45s | **.095** | .412s | .076 | 74min* | 1.25 | 10778.5X |
| HIGGS | .015 | .466s | .016 | .134s | .048 | 72.4s* | .012 | .644s | **.01** | .050s | .009 | 73min⋆ | 1.11 | 87149.2X |
| ijcnn | .032 | .353s | .030 | .105s | .032 | .018s | .027 | .313s | **.025** | **.006s** | .022 | 42min* | 1.14 | 759.6X |
| MNIST | .513 | 3.93s | .389 | 1.68s | .690 | 6.42s | .296 | 3.95s | **.290** | .234s | .270 | 20min* | 1.07 | 5067.5X |
| webspam | .047 | 1.00s | .043 | .414s | .061 | .641s | .020 | .756s | **.017** | **.031s** | .015 | 129s* | 1.13 | 4129.4X |
| bosch | .343 | 3.28s | .337 | 1.42s | .533 | 1.22s | .158 | 2.49s | **.143** | .213s | .100 | 237s* | 1.43 | 1112.0X |

| Robust GBDT | SignOPT | | HSJA | | RBA-Appr | | Cube | | LT-Attack (Ours) | | MILP | | Ours vs. MILP | |
|---|---|---|---|---|---|---|---|---|---|---|---|---|---|---|
| $\ell_2$ Perturbation | $\bar{r}$ | time | $\bar{r}$ | time | $\bar{r}$ | time | $\bar{r}$ | time | $\bar{r}_{our}$ | time | $r^*$ | time | $\bar{r}_{our}/r^*$ | Speedup |
| MNIST2-6 | 2.97 | 7.37s | 3.32 | 1.28s | 2.95 | **.156s** | 1.31 | 3.19s | **.971** | .438s | .762 | 25.0s* | 1.27 | 57.1X |
| breast-cancer | .437 | .711s | .449 | .069s | .436 | **.002s** | .940 | .239s | **.434** | .002s | .431 | .011s* | 1.01 | 5.2X |
| covtype | .076 | 1.11s | .104 | .196s | .137 | 3.26s | .096 | .726s | **.062** | **.047s** | .058 | 9min* | 1.07 | 11183.1X |
| diabetes | .142 | .591s | .150 | .061s | .161 | **.003s** | .274 | .240s | **.133** | .005s | .132 | .025s* | 1.01 | 4.8X |
| FMNIST | 1.67 | 9.27s | 1.34 | 1.64s | 3.72 | 7.01s | .500 | 4.45s | **.310** | .385s | .233 | 231s* | 1.33 | 600.8X |
| HIGGS | .020 | .879s | **.020** | .128s | .085 | 66.5s* | .023 | .580s | **.016** | .045s | .014 | 24min⋆ | 1.14 | 31715.5X |
| ijcnn | .033 | .572s | .035 | .096s | .040 | .014s | .042 | .307s | **.030** | **.006s** | .025 | .853s* | 1.20 | 140.3X |
| MNIST | 3.08 | 9.14s | 3.04 | 1.61s | 4.07 | 5.11s | 1.33 | 6.26s | **.932** | .291s | .670 | 7min* | 1.39 | 1523.6X |
| webspam | .097 | 3.24s | .100 | .431s | .148 | .589s | .068 | .869s | **.041** | **.034s** | .035 | 28.3s* | 1.17 | 840.6X |
| bosch | .750 | 9.62s | 2.33 | 1.54s | 1.45 | 1.21s | .480 | 3.84s | **.258** | .232s | .214 | 28.0s* | 1.21 | 120.7X |

Table B: Comparisons to prior works.

| | SignOPT | HSJA | Cube | RBA-Appr | **Ours** |
|---|---|---|---|---|---|
| Access Level | B-box | B-box | B-box | W-box + data | W-box |
| Search Space | input | input | input | training data | **leaf tuple** |
| Step Size | small $\eta$ | small $\xi$ | $\ell_0$ boundary | N/A | **one leaf node** |
| Queries / iter | 200 | 100∼632 | 100 | N/A | ∼**1** (line 203) |

Table C: RF statistics in addition to Table 7.

| Dataset | training set size | test set size | subsample | acc. |
|---|---|---|---|---|
| MNIST2-6 | 11,876 | 1,990 | .8 | .963 |
| diabetes | 614 | 154 | .8 | .775 |
| FMNIST | 60,000 | 10,000 | .8 | .823 |
| higgs | 10,500,000 | 500,000 | .8 | .702 |
| ijcnn | 49,990 | 91,701 | .8 | .919 |
| bosch | 946,997 | 236,750 | .8 | .994 |

Table D: RF results in addition to Table 8.

| | Cube | | Ours | | MILP | | Ours vs. MILP | |
|---|---|---|---|---|---|---|---|---|
| $\ell_2$ Perturbation | $\bar{r}$ | time | $\bar{r}_{our}$ | time | $r^*$ | time | $\bar{r}_{our}/r^*$ | Speedup |
| MNIST2-6 | .439 | 2.13s | .207 | .045s | .194 | .071s | 1.07 | 1.6X |
| diabetes | .260 | .285s | .151 | .003s | .146 | .042s | 1.03 | 14.X |
| FMNIST | .141 | 3.51s | .066 | .080s | .066 | 7.44s | 1.00 | 93.X |
| higgs | .015 | .423s | .009 | .013s | .009 | 6.66s | 1.00 | 512.3X |
| ijcnn | .046 | .336s | .028 | .003s | .028 | .185s | 1.00 | 61.7X |

**[R3] 1) Challenging datasets:** In Table 2 and 3, HIGGS contains 10.5 million training examples and the ensemble
has 300 trees. We additionally added Bosch (1.2 million examples, 968 features) in Table A. Both datasets are from
challenging Kaggle competitions. Our method is effective on both datasets. **2) C++/Python:** Among the baselines,
we implemented RBA-Appr in C++. MILP uses a thin wrapper around the Gurobi Solver. Other methods spend
majority of time on XGBoost model inference rather than Python code. For instance, on Fashion-MNIST, SignOPT,
HSJA, Cube spent 72.8%, 57.3%, 73.4% of runtime in XGBoost library (C++) calls, respectively. **3) Ablation**
**experiments:** Our ablation experiments are spread across the paper: **(a)** *Size of the neighborhood:* we compare
the effect of small (NaiveFeature) and large (NaiveLeaf) neighborhood space in Table 1, and study the minimum
neighborhood distance in Appendix D.3. **(b)** *Random noise optimization* also improves the solution quality. We provide
baseline results in Table 1 and optimized results in Table 2 and 3. **(c)** *number of initial examples* affects both the
runtime and the solution quality, and we compare the effect in Figure 2. **4) Bounding boxes:** The exact definition is
$B(\mathcal{C}) = \bigcap_{i\in\mathcal{C}} B^i = \bigcap_{i\in\mathcal{C}}[l_1^i, r_1^i] \times \cdots \times \bigcap_{i\in\mathcal{C}}[l_d^i, r_d^i]$. It is the Cartesian product of the intersection on each feature
dimension. **5) Why $x'$ and $a$ in Figure 1 are local minimums:** Decision-based attacks update solution along the
decision boundary. They will be trapped at $x'$ and $a$ since small perturbation on both sides will increase the distance to
$x_0$. To update from $a$ to $b$, the path will be $a \to (5, 10) \to b$, but since $a \to (5, 10)$ will increase the distortion they
won't find this path. Other methods such as random sampling is inefficient in a large $\ell_p$ ball in the order of $\|a - b\|_p$.

**[R4] 1) Motivation of minimizing $\ell_p$ perturbation:** We minimize the perturbation to find a *smallest possible* attack,
to uncover the true weakness of a model. $\ell_p$ distance is widely used in previous attacks (Carlini, Wagner, 2017;
Kantchelian et al., 2015) and its prevalence is mostly due to mathematical convenience. Small $\ell_p$ perturbations are
usually invisible, but we agree it cannot capture many real settings. Our method can be adapted to other distance metrics:
in line 8 of Alg. 1, we enumerate the distances between $x_0$ and a set of candidates $\mathcal{C}$ to find the minimum. This distance
can be redefined. **2) Distance notation:** We will clean up notation and use $\text{dist}_p(\mathcal{C}, x_0)$ to denote the $\ell_p$ distance.

**[R5] 1) Size of neighborhood:** Thanks for the correct understanding on this trade-off. Our ablation (Table 1) and
experiments (Table 2, 3) empirically show that distance 1 is sufficient for outperforming other attacks. **2) Robust to**
**structure changes:** For each tree, its non-leaf nodes and structures are irrelevant to our algorithm as long as the leaves
produce the same bounding boxes. We conduct a small experiment on adversarial training and improve the $\ell_2$ robustness
from .082 to .115 on diabetes dataset. **3) Random forest:** We added the remaining experiments in Table D and C.

[Meta-Review · NeurIPS 2020]

This paper presents a more efficient method for finding adversarial examples for tree ensembles -- that is, finding an instance with a different predicted label that is close to a given query instance. It's good to see work on adversarial attacks against tree ensembles, since they're widely used but rarely studied. This is a nice step forward.